# Energy-efficient pathway for selectively exciting solute molecules to high vibrational states via solvent vibration-polariton pumping

Tao E. Li 📧 [1,2], Abraham Nitzan[1,3] 📧 & Joseph E. Subotnik 📧 [1]

Selectively exciting target molecules to high vibrational states is inefficient in the liquid phase, which restricts the use of IR pumping to catalyze ground-state chemical reactions. Here, we demonstrate that this inefficiency can sometimes be solved by confining the liquid to an optical cavity under vibrational strong coupling conditions. For a liquid solution of $^{13}CO_2$ solute in a $^{12}CO_2$ solvent, cavity molecular dynamics simulations show that exciting a polariton (hybrid light-matter state) of the solvent with an intense laser pulse, under suitable resonant conditions, may lead to a very strong (>3 quanta) and ultrafast (<1 ps) excitation of the solute, even though the solvent ends up being barely excited. By contrast, outside a cavity the same input pulse fluence can excite the solute by only half a vibrational quantum and the selectivity of excitation is low. Our finding is robust under different cavity volumes, which may lead to observable cavity enhancement on IR photochemical reactions in Fabry–Pérot cavities.

---

[1] Department of Chemistry, University of Pennsylvania, Philadelphia, PA 19104, USA. [2] Department of Chemistry, Yale University, New Haven, CT 06520, USA. [3] School of Chemistry, Tel Aviv University, Tel Aviv 69978, Israel. 📧email: taoli@sas.upenn.edu; anitzan@sas.upenn.edu; subotnik@sas.upenn.edu

Controlling (electronic) ground-state chemical reaction rates with an external infrared (IR) laser is a long-standing goal of IR photochemistry. Although it is possible to trigger chemical reactions by exciting the reactants to high vibrational excited states via ladder climbing[1], such IR-controlled chemical reactions are rarely efficient in the liquid phase because of competing relaxation processes that transfer vibrational energy to other degrees of freedom, either internal modes of the molecule or external modes of surrounding molecules (leading to a build-up of heat in the environment)[2]. Up to now, there have been only a few observations of selective IR photochemistry (i.e. reactions where an IR pump promotes selective reactions beyond a simple heating effect)[3–5]. That being said, if there were a general mechanism for targeting some molecules (e.g., reactants or solute molecules) and achieving highly vibrationally excited states on a timescale shorter than the lifetime of vibrational energy relaxation, one could imagine using IR light to help catalyze reactions of interest.

During the past decade, a novel means to control molecular properties has emerged through the formation of strong light–matter coupling between a vacuum electromagnetic field and a molecular (electronic or vibrational) transition of a large ensemble of molecules[6–9]. Under this collective strong light–matter coupling regime, the formed hybrid light–matter states, known as molecular polaritons, can suppress ultraviolet–visible (UV–vis) photochemical reactions[10], alter ground-state reaction rates[11–13], promote electronic conductivity[14], and facilitate energy transfer between different molecular species[15–20], etc. For example, under vibrational strong coupling (VSC)[21,22] between an optical cavity mode and an equal liquid mixture of $W(^{12}CO)_6$ and $W(^{13}CO)_6$, Xiang et al. [20] have experimentally demonstrated that, by pumping the upper vibrational polariton (UP) with an external IR laser, intermolecular vibrational energy transfer between the two molecular species can be accelerated to a lifetime of tens of ps due to the intrinsically delocalized nature of polaritons.

Apart from the acceleration of energy transfer due to polariton delocalization, the other intriguing consequence of VSC is the enhancement of molecular nonlinear absorption[7,23–25]. Due to the anharmonicity of molecular vibrations, it is possible to tune a cavity in such a way that twice the lower polariton (LP) energy roughly matches the molecular $0 \rightarrow 2$ vibrational transition; in such a case, pumping the LP can facilitate molecular nonlinear absorption and promote the creation of highly vibrationally excited molecules[7,24].

Here, we will numerically demonstrate that by combining the above two unique features of VSC—polariton delocalization and molecular anharmonicity—it is possible to selectively excite a set of molecules, i.e., a small concentration of solute molecules, to highly vibrationally excited states and leave the vast majority of solvent molecules barely excited via strongly exciting the solvent LP. We emphasize that this finding is of crucial importance, as it lays the foundations for using IR polaritonic chemistry to promote chemical reaction rates. Moreover, because the solvent LP is essentially composed of only solvent modes plus the cavity mode (with a negligible contribution from solute molecules), the observation of energy accumulation in the solute molecules must be regarded as vibrational energy transfer from solvent to solute, which is entropically unfavorable outside the cavity. The difference between our finding and conventional multiphoton molecular nonlinear absorption is also discussed in Supplementary Figs. 2, 3, and 8.

While many theoretical approaches[26–33] have been developed recently in order to describe molecular polaritons, our numerical approach is classical cavity molecular dynamics (CavMD) simulation[24,28,34], a new and promising tool for propagating coupled photon–nuclear dynamics in the condensed phase, though with the limitations of using a classical force field to describe realistic molecules; see Fig. 1 for the simulation setup and the section "Methods" (and also Supplementary Methods) for more details.

## Results

**Polaritonic energy transfer in pure liquid.** Before presenting the main finding of this manuscript—selective energy transfer to solute molecules—we investigate how exciting a polariton leads to a nonuniform energy redistribution in a pure liquid $^{12}CO_2$ system (with $N_{sub} = 216$ molecules explicitly propagated in a simulation cell) at room temperature (300 K). In Fig. 2a, we plot the equilibrium IR spectrum outside a cavity (black line) or inside a cavity (red line), where the IR spectrum is calculated by evaluating the auto-correlation function of the total dipole moment for the molecular subsystem[24,35]. Inside the cavity, the C=O asymmetric stretch mode (peaked at $\omega_0 = 2327$ cm$^{-1}$) is resonantly coupled to a cavity mode (at $\omega_c = 2320$ cm$^{-1}$ corresponding to the vertical blue line) with an effective coupling strength $\widetilde{\varepsilon} = 2 \times 10^{-4}$ a.u. In this collective VSC regime, a pair of polaritons [LP ($\omega_{LP} = 2241$ cm$^{-1}$) and UP ($\omega_{UP} = 2428$ cm$^{-1}$)] forms and these two peaks are separated by a Rabi splitting of 187 cm$^{-1}$.

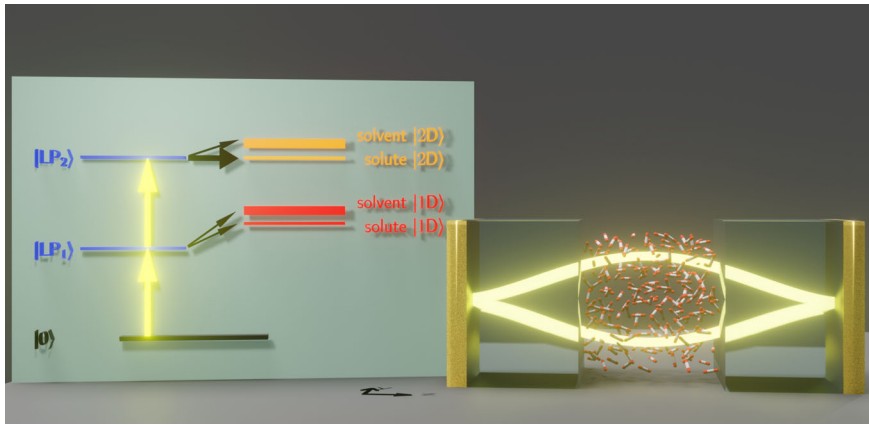

**Fig. 1 Sketch of the cavity setup for CavMD simulations.** $N_{sub}$ molecules are confined in a cavity formed by a pair of parallel metallic mirrors. These molecules are coupled to a single cavity mode (with two polarization directions) and are simulated in a periodic simulation cell. Our classical CavMD simulations show that, after a strong excitation of the solvent lower polariton (LP) with an IR laser pulse, the input energy can be selectively transferred to the highly excited dark states of the solute molecules, leaving the vast majority of the solvent molecules barely excited. The left cartoon demonstrates the quantum analog of our classical finding; see Supplementary Note 8 for details.

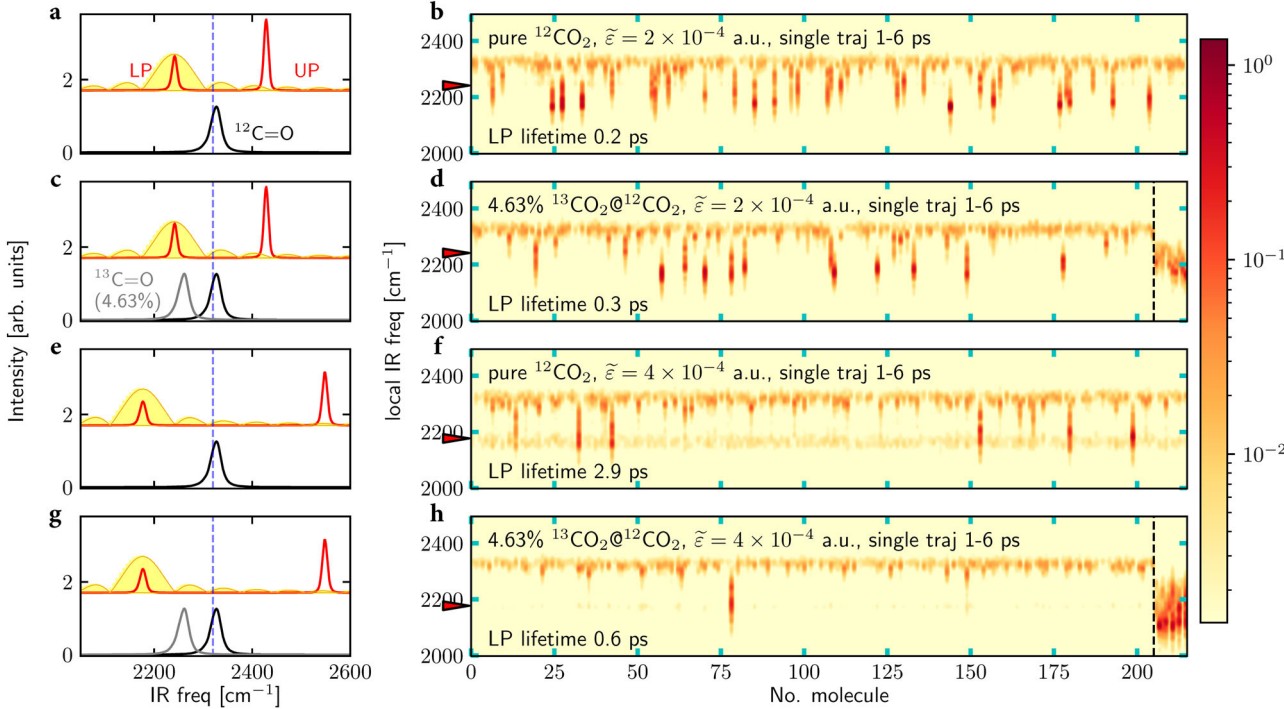

**Fig. 2 Rabi splitting and selective polariton energy transfer. a** Equilibrium IR spectrum for pure liquid $^{12}CO_2$ outside a cavity (black line) or inside a cavity with coupling $\widetilde{\varepsilon} = 2 \times 10^{-4}$ a.u. (red line); the vertical dashed blue line denotes the cavity mode at 2320 cm$^{-1}$. **b** The corresponding nonequilibrium local IR spectrum for each molecule after a resonant excitation of the LP at 2241 cm$^{-1}$ with a strong external pulse. The external pulse takes the form $E_0 \cos(\omega t + \phi)\mathbf{e}_x$ (0.1 < t < 0.6 ps) with fluence 632 mJ/cm$^2$; the corresponding pulse spectrum is also plotted as the yellow curve on the left panel. The nonequilibrium spectrum is calculated by evaluating the dipole autocorrelation function of each $CO_2$ molecule over time interval 1–6 ps. The red arrow marks the frequency of the LP and also the external pulse, and the LP lifetime (0.2 ps, calculated from photonic energy dynamics[24]) is labeled at the bottom. **c** Equilibrium IR spectrum (red line) for 4.63% $^{13}CO_2$ in a $^{12}CO_2$ solution when $\widetilde{\varepsilon} = 2 \times 10^{-4}$ a.u. The gray line denotes the $^{13}CO_2$ IR spectrum outside the cavity; again, the black line denotes the pure $^{12}CO_2$ IR spectrum outside the cavity. **d** The corresponding nonequilibrium local IR spectrum for each molecule plotted in a similar manner as (**b**). The responses of $^{12}CO_2$ (left) and $^{13}CO_2$ (right) are separated by the vertical dashed line. **e–h** The same plots as in (**a–d**) but with a larger effective light-matter coupling ($\widetilde{\varepsilon} = 4 \times 10^{-4}$ a.u.). Note that with an appropriate LP frequency, the LP is more likely to transfer energy to the $^{13}CO_2$ solute molecules than to the $^{12}CO_2$ solvent. See Supplementary Methods for simulation details, the definition of the $CO_2$ force field, and the definition of the spectra signals.

We now excite the LP with a strong pulse of the form $\mathbf{E}(t) = E_0 \cos(\omega t)\mathbf{e}_x$ (0.1 < t < 0.6 ps) with fluence 632 mJ/cm$^2$ ($E_0 = 6 \times 10^{-3}$ a.u.); see the yellow curve in Fig. 2a for the corresponding pulse spectrum. This fluence will be used below and throughout the present manuscript, and the pulse is assumed to interact only with the molecular subsystem. Note that, as shown in Supplementary Fig. 6, for the same amount of pumped energy, using a Gaussian pulse (instead of a square pulse) does not meaningfully alter the reported results; similarly, as shown in Supplementary Fig. 5, if we assume that the external pulse activates the cavity mode directly (rather than the molecular subsystem), the reported results are effectively unchanged.

In Fig. 2b, after the LP pumping, the resulting energy distribution among molecules is characterized by calculating the response of every $^{12}CO_2$ molecule in the simulation cell after the LP excitation. The individual molecular response is calculated by evaluating the auto-correlation function of each single-molecule dipole moment (which we call the "local" IR spectrum)[24] during a time window 1 < t < 6 ps (immediately after the LP excitation). Note that the response measured is different from the usual IR spectrum—which is calculated by evaluating the auto-correlation function of the total dipole moment of a molecular system (as in Fig. 2a) and reflects the dynamics of the molecular bright mode; the local IR spectrum measured here reflects the dynamics of individual molecules and can be expressed as a linear combination of molecular bright and dark modes. Because the total

number of molecules explicitly accounted for in the simulation is large (with $N_{sub} = 216$), the local IR spectrum is dominated by the density of dark modes.

In Fig. 2b, every column of pixels represents the local IR spectrum of one molecule (in total $N_{sub} = 216$ molecules) during 1–6 ps and the color bar (from light yellow to deep red) denotes a logarithmic scaled spectroscopic intensity. Clearly, after the LP excitation (where the LP lifetime fitted from photonic energy dynamics[24] is 0.2 ps), the polaritonic energy is nonuniformly transferred to different $^{12}CO_2$ molecules (which are composed mostly of vibrational dark modes). While most molecules are only weakly excited and have a peak near 2320 cm$^{-1}$ (the equilibrium IR peak; see also Fig. 2a), a dozen of molecules are strongly excited and show an intense, red-shifted peak near 2200 cm$^{-1}$. Because we use an anharmonic force field[24] to simulate carbon dioxide molecules, the presence of intense, red-shifted peaks implies that the LP energy is mostly transferred to a small fraction of $^{12}CO_2$ molecules.

The nonuniform polaritonic energy redistribution in Fig. 2b stems from polariton-enhanced molecular nonlinear absorption, a mechanism which has been shown experimentally (via two-dimensional IR spectroscopy)[23], analytically[25], and numerically (via CavMD)[24]. Quantum-mechanically speaking, when twice the LP energy roughly matches the $0 \rightarrow 2$ vibrational transition of molecules, the LP can serve as a "virtual state" to enhance molecular nonlinear absorption of light and directly create highly

vibrational excited molecules within a sub-ps timescale. In the present case (see Fig. 2a), the LP frequency sits between the fundamental $^{12}C=O$ asymmetric stretch ($\omega_0 = 2327$ cm$^{-1}$) and the $1 \rightarrow 2$ vibrational transition (which is roughly 2200 cm$^{-1}$). Hence, polariton-enhanced molecular nonlinear absorption can occur once the LP is strongly excited. Moreover, since molecules have different orientations and different local electrostatic environments (which leads to different instantaneous vibrational frequencies), only a small subset of molecules can interact strongly with the LP through the nonlinear channel (which requires an exact frequency match), thus leading to a vast difference in energy redistribution among molecules. In the end, some solvent molecules become highly excited and others do not. This quantum-mechanical interpretation of polariton-enhanced molecular nonlinear absorption is easy to understand. A classical analog can be pictured as well. Once the LP is excited and starts to dephase, due to the inhomogeneous local molecular environment, some molecules receive more polaritonic energy than others. Due to molecular anharmonicity, these high-energy molecules oscillate with red-shifted, closer-to-LP frequencies, leading to an even stronger interaction between these high-energy molecules and the LP. Such "self-catalyzing" can, under strong LP excitation, eventually lead to some molecules being strongly and nonlinearly excited. While keeping in mind that our calculations are in fact classical, below we will use the quantum description because we feel the latter is more standard in the literature.

**Polaritonic energy transfer to solute molecules**. Let us now show that the mechanism above can be utilized to selectively transfer the energy from an IR laser pulse to a few target molecules in a liquid system. For the sake of simplicity, let us choose these target molecules to be a few $^{13}CO_2$ molecules and investigate how to achieve selective energy transfer to these $^{13}CO_2$ solute molecules (which are dissolved in a liquid $^{12}CO_2$ solution).

For a solution of 4.63% (10/216) $^{13}CO_2$ molecules dissolved in liquid $^{12}CO_2$ (in total $N_{sub} = 216$ molecules are included in the simulation cell), the red line in Fig. 2c plots the equilibrium IR spectrum inside the same cavity as in Fig. 2a, b (with $\omega_c = 2320$ cm$^{-1}$ and $\tilde{\varepsilon} = 2 \times 10^{-4}$ a.u.). Because the concentration of the $^{13}CO_2$ solute is small, inside the cavity, the positions of the polaritons are largely unchanged compared with the case of pure liquid $^{12}CO_2$ (in Fig. 2a). Note that the equilibrium IR spectrum for pure liquid $^{13}CO_2$ system outside a cavity is also plotted as the gray line in Fig. 2c, where the $^{13}C=O$ asymmetric stretch peaks at 2262 cm$^{-1}$.

For this liquid mixture, after a strong excitation of the LP (again with a pulse fluence of 632 mJ/cm$^2$), Fig. 2d plots the transient local IR spectra for every $^{12}CO_2$ and $^{13}CO_2$ molecule. Here, the $^{12}CO_2$ are plotted on the left and the $^{13}CO_2$ molecules are plotted on the right-hand side of the vertical dashed line. Figure 2d shows that there is now a competition between exciting $^{12}CO_2$ and $^{13}CO_2$ molecules, as molecules of both isotopes exhibit intense, red-shifted peaks.

This conclusion is illustrated quantitatively in Fig. 3a, where we plot the dynamics of the average C=O bond potential energy (minus the thermal energy $k_BT$) per $^{12}CO_2$ (gray line) or $^{13}CO_2$ (red line) molecule after the strong LP excitation. Due to the LP excitation [where the time window of the pulse (0.1 < $t$ < 0.6 ps) is represented with a yellow region], the vibrational energy of the $^{13}CO_2$ solute molecules can reach roughly four times that of the $^{12}CO_2$ solvent molecules and each solute molecule can absorb roughly a vibrational quantum (~2 × 10$^3$ cm$^{-1}$) of the input energy. Note that Fig. 3a is drawn with a logarithmic scale; see the black arrow which denotes the energy difference. By contrast, outside the cavity, when the $^{13}C=O$ asymmetric stretch of the liquid mixture

is resonantly excited by the same pulse (Fig. 3c), the $^{13}CO_2$ vibrational energy reaches roughly twice the energy of the $^{12}CO_2$ solvent molecules, and the energy absorption of each solute molecule is roughly half a vibrational quantum (~10$^3$ cm$^{-1}$). Our finding indicates that the cavity environment can meaningfully improve the selectivity and also the absolute value of solute excitation by a factor of two (though a factor of two is perhaps not a huge effect).

**Improving the selectivity of energy transfer to solute molecules by tuning the Rabi splitting**. In order to improve the selectivity of energy transfer from the LP, we need to both (i) suppress the energy transfer to the $^{12}CO_2$ solvent molecules and also (ii) enhance the energy transfer to the $^{13}CO_2$ solute molecules. As mentioned above, the nonuniform polaritonic energy transfer stems from polariton-enhanced molecular nonlinear absorption, a mechanism that requires twice the LP frequency to roughly match the $0 \rightarrow 2$ vibrational transition. In other words, by changing the LP frequency relative to the vibration of one molecular species, we can enhance or suppress polaritonic energy transfer to that molecular species according to a frequency match or mismatch.

Let us first focus on the side of the $^{12}CO_2$ molecules and investigate how to suppress the energy transfer to the $^{12}CO_2$ molecules with an increased Rabi splitting. For a pure liquid $^{12}CO_2$ system, when the effective light–matter coupling strength is increased from $\tilde{\varepsilon} = 2 \times 10^{-4}$ a.u. to $\tilde{\varepsilon} = 4 \times 10^{-4}$ a.u., the red line in Fig. 2e plots the equilibrium IR spectrum inside the cavity with $\omega_c = 2320$ cm$^{-1}$. For the larger light–matter coupling ($\tilde{\varepsilon} = 4 \times 10^{-4}$ a.u.), the Rabi splitting increases by a factor of two and the LP frequency red-shifts from 2241 cm$^{-1}$ (in Fig. 2a–d) to 2177 cm$^{-1}$. Now, as shown in Fig. 2f, after resonant excitation of the LP with a strong laser pulse, polaritonic energy transfer to $^{12}CO_2$ molecules is largely suppressed compared with Fig. 2b. The suppression of polaritonic energy transfer is consistent with a longer LP lifetime (2.9 ps) relative to that in Fig. 2b (0.2 ps); this longer LP lifetime can be observed in Fig. 2f by visualizing the near-uniform intensity among all molecules at the LP frequency (positioned by the red arrow) over the time scale 1–6 ps, which corresponds to a long-lived molecular bright state.

Second, consider the case where the liquid mixture of 4.63% $^{13}CO_2$ in a $^{12}CO_2$ solution experiences a larger light–matter coupling (with $\tilde{\varepsilon} = 4 \times 10^{-4}$ a.u.). Because the concentration of the $^{13}CO_2$ solute molecules is small, the equilibrium IR spectrum inside the cavity (Fig. 2g) remains unchanged compared with the case of pure liquid $^{12}CO_2$ (Fig. 2e). After resonantly exciting the LP with a strong laser pulse, as shown in Fig. 2h, we find that the polaritonic energy mostly transfers to the $^{13}CO_2$ molecules (on the right-hand side of the vertical dashed white line). Quantitatively speaking, Fig. 3b plots the corresponding dynamics of the average C=O bond potential energy for the solute and solvent molecules. Here, within the time window of the laser pulse ($t < 1$ ps), the solute molecules can be excited to a state of roughly three vibrational quanta (recall that the frequency of the C=O asymmetric stretch is rough ~2 × 10$^3$ cm$^{-1}$), which is six times of the outside-cavity solute energy absorption (Fig. 3c). As far as selectivity is considered, within $t < 5$ ps, the solute molecules absorb an energy 56 times larger than the solvent molecules (see the black arrow), a difference which is one order of magnitude larger than the case of outside the cavity (Fig. 3c). In short, although we found above that, when $\tilde{\varepsilon} = 2 \times 10^{-4}$ a.u., there is a competition between the polaritonic energy transfer to $^{12}CO_2$ and $^{13}CO_2$ molecules, we now find that when $\tilde{\varepsilon} = 4 \times 10^{-4}$ a.u., $^{13}CO_2$ clearly wins.

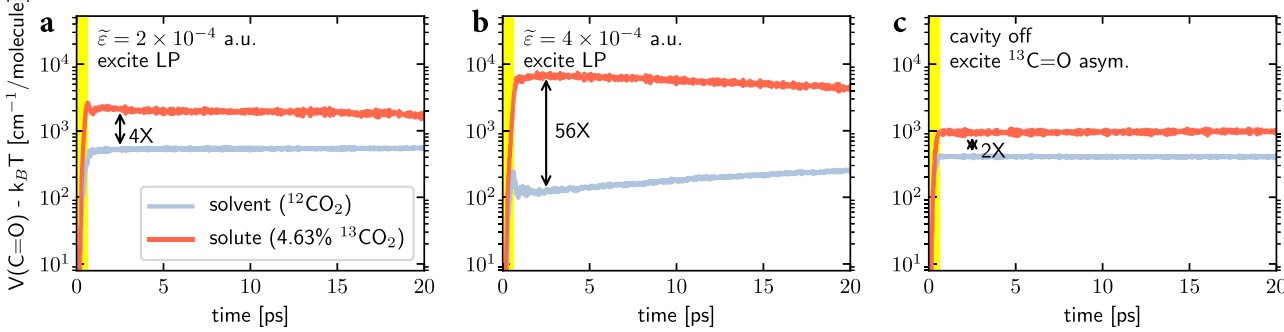

**Fig. 3 Time-resolved dynamics for the average C=O bond potential energy per molecule.** Three conditions are compared: (**a**) $\widetilde{\varepsilon} = 2 \times 10^{-4}$ a.u. after exciting the LP at 2241 cm$^{-1}$ (the same as Fig. 2d); (**b**) $\widetilde{\varepsilon} = 4 \times 10^{-4}$ a.u. after exciting the LP at 2177 cm$^{-1}$ (the same as Fig. 2h); and (**c**) outside the cavity after exciting the $^{13}$C=O asymmetric stretch at 2262 cm$^{-1}$. In each subplot, the $^{12}CO_2$ solvent is drawn with a gray line and the 4.63% $^{13}CO_2$ solute is drawn with a red line. The y-axis is plotted in a logarithmic scale, and the yellow region denotes the time window (0.1 < t < 0.6 ps) during which the external pulse is applied. Note that although the nonequilibrium local IR spectrum in Fig. 2 was calculated using a single trajectory, 40 trajectories are averaged here to obtain an ensemble-averaged result; see Supplementary Methods for simulation details.

The high selectivity of this polaritonic energy transfer to the solute molecules can be explained as follows. First, since the LP frequency is unchanged compared with the pure liquid $^{12}CO_2$ system in Fig. 2e, polariton-enhanced molecular nonlinear absorption for the $^{12}CO_2$ solvent molecules remains greatly suppressed. Second, for the larger Rabi splitting, the vibrational frequency difference between the LP and $^{13}$C=O asymmetric stretch (see red and gray lines in Fig. 2g) is suitable to allow for polariton-enhanced molecular nonlinear absorption of the solute molecules. This statement is consistent with the shortened LP lifetime in Fig. 2h (0.6 ps) as compared with $\tau_{LP} = 2.9$ ps in Fig. 2f. Altogether, these two facts lead to a high selectivity in energy transfer. A summary of the quantum picture of our finding—a selective polariton energy transfer to the second excited solute dark states when twice the solvent LP frequency roughly matches the solute $0 \rightarrow 2$ transition—is plotted in cartoon form on the left-hand side of Fig. 1; see also Supplementary Note 8 for details.

**Dependence on system size and solute concentration.** One important issue we must address is the dependence of polaritonic energy transfer on molecular system size. This study is necessary because usual VSC setups in experiments (e.g., Fabry–Pérot microcavities) contain a macroscopic number of molecules, while we have only simulated a molecular system of $N_{sub} = 216$ coupled to the cavity. Therefore, we must check how our conclusions vis a vis VSC effects depend on molecular system size.

Following the approach from ref. [24], in Fig. 3b, we enlarge the molecular system by simultaneously increasing both the number of $^{13}CO_2$ and $^{12}CO_2$ molecules (maintaining a constant proportion) and we decrease the light–matter coupling strength ($\widetilde{\varepsilon}$); these two effects are chosen in a balanced manner ($\widetilde{\varepsilon} \propto 1/\sqrt{N_{sub}}$) so as to keep the molecular density and Rabi splitting the same as in Fig. 2g, h. For systems with different sizes, after the same strong excitation of the LP, Fig. 4 plots the dynamics of the average C=O bond potential energy for the $^{13}CO_2$ and $^{12}CO_2$ molecules. Here, the top and bottom lines denote the dynamics of $^{13}CO_2$ and $^{12}CO_2$ molecules, respectively. Lines with different colors (from black to orange) denote molecular systems of different sizes (from $N_{sub} = 216$ to $N_{sub} = 216 \times 16$). Note that after the pulse, the $^{13}CO_2$ molecules are hot and start to cool down, while the weakly excited $^{12}CO_2$ molecules start to heat up. Figure 4 also clearly shows that, although the later-time vibrational relaxation and energy transfer dynamics (t > 5 ps) can depend on the size of the molecular system[34], we do

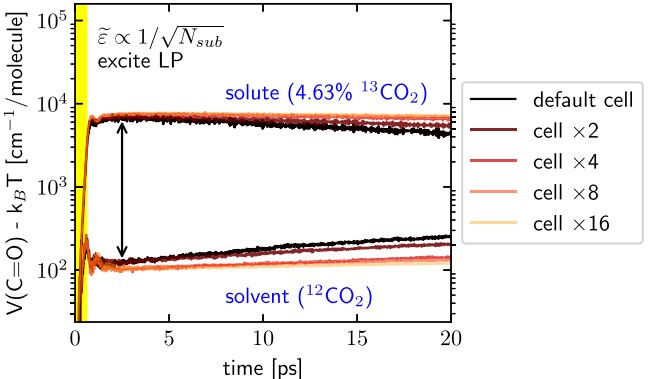

**Fig. 4 Effects of the molecular system size on the dynamics of the C=O bond potential energy at constant solute concentration.** The Rabi splitting and molecular number density have been fixed. Black lines correspond to the data in Fig. 3b ($N_{sub} = 216$), and the lines from dark red to orange represent increasing molecular systems where $N_{sub}$ is increased by a factor of $N$ ($N = 2, 4, 8, 16$). In all cases, we excite the LP and measure the amount of potential energy in the C=O bonds. Note that the selective polaritonic energy transfer to the $^{13}CO_2$ solute molecules is fairly robust against molecular system size, indicating that the current finding may hold for collective VSC with large cavity volumes but under the same Rabi splitting, i.e., smaller $\widetilde{\varepsilon}$; see text below and also Supplementary Fig. 1 for an explanation of the system size dependence.

observe a consistent selective polaritonic energy transfer behavior at early times (t < 5 ps) for all molecular system sizes. Therefore, we tentatively conclude that our numerical findings here should be observable in cavities with different volumes, including Fabry–Pérot microcavities (that are usually used to study collective VSC) and plasmonic cavities, an emerging platform for studying VSC[36,37].

Note that the robustness of the early-time (<1 ps) polaritonic decay dynamics against the system size can be rationalized by, e.g., Fermi's golden rule calculations. Although the interaction between the polariton and each dark mode scales with $1/N$ (where $N$ denotes the total molecular number) via intermolecular interactions, since there are $N-1$ dark modes that allow polaritonic energy transfer (or dephasing), when Fermi's golden rule is used to describe the polaritonic relaxation dynamics, the $1/N$ and $N-1$ factors cancel with each other, leading to consistent early-time dynamics during polaritonic relaxation versus the system size. This argument explains why the early time dynamics (<1 ps) in Fig. 4 do not depend sensitively on $N$. That being said, for the

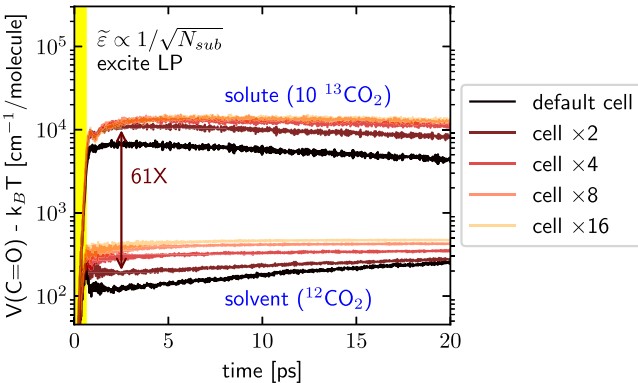

**Fig. 5 Effects of the molecular system size on the dynamics of the C=O bond potential energy at constant solute molecular number.** This plot is the same as Fig. 4 except for that the $^{13}CO_2$ solute molecular number is fixed as 10 when the system size enlarges. Note that, when the system size is increased (or when the solute concentration is decreased), both the solvent and solute molecules are more heavily pumped under the same IR excitation of the LP. See text for details of this counter-intuitive feature.

later-time (>5 ps) dynamics, which involve energy transfer and vibrational relaxation, the same argument shows that the presence of a cavity will play a role and lead to $N$-dependent dynamics. After all, consider the following energy transfer channel that we might expect to be important after we have excited a set of $^{13}CO_2$ molecules: $^{13}CO_2 \rightarrow$ polaritons $\rightarrow$ $^{12}CO_2$. Since there are only two (not $N$) polaritons, we do not expect this pathway to play an important role once the system size is very large, leading to slower vibrational relaxation and energy transfer dynamics in later times; see ref. [34] for a detailed study.

Finally, in Fig. 5, we plot the dynamics of the averaged C=O bond potential energy per $^{13}CO_2$ solute or $^{12}CO_2$ solvent molecule as a function of time for different system sizes—but now with a fixed number (10) of solute molecules. In other words, we increase only the number of $^{12}CO_2$ solvent molecules (so that the solute concentration is decreased). For all curves, as above, the total Rabi splitting is kept constant by decreasing the effective coupling strength $\widetilde{\varepsilon}$, which corresponds physically to placing the system in a larger volume cavity.

Perhaps counter-intuitively, Fig. 5 shows that the C=O bond energy of both the solvent and solute molecules increases as the number of solvent molecules increases. These findings can be explained as follows. On the one hand, as for the solute molecules, when the number of solvent molecules increases, the total transition dipole of the solvent bright mode also increases, such that the total amount of energy absorbed through the polariton grows when the same incident pulse is applied—and each solute molecule will recover more energy from the polariton through the nonlinear channel (since the energy absorbed by the polariton divided by the solute number grows). On the other hand, when the concentration of the solute decreases (under larger system sizes), the pumped solvent polariton cannot efficiently transfer energy to its favorite acceptor (the solute), such that the polariton transfers more energy to the solvent molecules. Finally, with regards to selectivity (i.e., energy of solute divided by energy of solvent), these two effects clearly work against each other. Empirically, we find that the selectivity reaches a maximum value of 61 times (see the vertical brown arrow) and the solute excitation can exceed $10^4$ cm$^{-1}$ (4–5 quanta) when we double the system size (i.e. we add slightly more than two times the number of solvent molecules so that the solute concentration becomes 2.32%, the brown lines). Very interestingly, in Supplementary Fig. 7 we

show that the selectivity also goes through a maximum value as a function of the pulse fluence, confirming our assertion that energy transfer to the $^{13}CO_2$ species is dominated by a nonlinear transition: at low intensities, this mechanism is inefficient, while at high intensities the $^{13}CO_2$ species energy absorption becomes saturated.

The findings above cannot be achieved outside a cavity. For example, on the one hand, when the solvent bright mode is pumped outside a cavity, since the excited bright mode dephases on a much faster timescale than the polaritons[38], and because there are no resonance conditions between the solvent bright mode and the solute, energy transfer from the solvent to solute is not efficient and occurs on a much longer timescale than 1 ps. On the other hand, when the solute vibration is directly excited outside the cavity (see Fig. 3c), the energy absorbed by the system is much smaller than the inside-cavity case (where energy absorption is dominated by the solvent polariton).

## Discussion

To summarize, we have numerically demonstrated that energy from a strong IR laser pulse can be selectively transferred to solute molecules by pumping the LP of the solvent molecules. This selective energy transfer behavior requires the LP to have an appropriate frequency to selectively enhance molecular nonlinear absorption for the solute (but not solvent) molecules and can be optimized by changing the solute concentration. Consequently, the transient vibrational energy difference between the solute and the solvent molecules can exceed the free-space case by more than one order of magnitude. Since our CavMD simulations have shown some key agreements with VSC experiments (including the signature of polariton-enhanced molecular nonlinear absorption), and because the current finding of selective energy transfer to the solute molecules is robust against the cavity volume, we believe that this finding is also amenable to experimental verification. For example, although the molecule-resolved nonequilibrium local IR spectrum (see Fig. 2) cannot be directly observed experimentally, for Fabry–Pérot microcavities, pump-probe or 2D-IR spectroscopy should be able to verify the main predictions of this paper. For example, once a small fraction of solute molecules is added, the observation of a strong reduction of the solvent LP lifetime when twice the solvent LP frequency roughly matches the solute $0 \rightarrow 2$ transition would provide one indirect piece of evidence supporting our mechanism. Moreover, for certain experiments, after pumping the solvent LP, the observation of bond dissociation events would serve as a direct piece of evidence confirming that solute molecules have been highly excited.

In principle, our finding might be one route to IR laser-controlled chemistry on an electronic ground-state surface, which remains difficult to achieve in the liquid phase. In order to selectively excite the (presumably chemically active) solute molecules in an efficient way, it would be best if the solvent molecules were to have a relatively simple vibrational structure so that the dephasing channel from the solvent polariton to the solvent dark modes can be largely forbidden. For the solute molecules, however, a complicated vibrational density of states might actually enhance the energy transfer from the solvent polariton. One must wonder if one potential application of the present manuscript would be isotope separation, where two similar molecules can be distinguished by a vibrational degree of freedom. Finally, for the cases where the vibrational relaxation is so strong that the transient excitation of the solute molecules would not facilitate IR photochemistry, the current mechanism may also allow us to achieve selective heating of the local geometry, e.g., by putting the solute molecules inside a protein,

perhaps one can control a selective denaturation of proteins without heating up the whole liquid sample. Overall, there are clearly many future possibilities for using collective VSC to modify molecular properties.

## Methods

The CavMD approach is formulated in the dipole gauge and includes the self-dipole term in the light–matter Hamiltonian, thus preserving gauge invariance[39–41]. The motivation for using classical CavMD for simulating nonequilibrium vibrational dynamics inside the cavity is twofold. First, for molecular vibrational relaxation and energy transfer dynamics in the condensed phase and outside the cavity, classical (nonequilibrium) molecular dynamics simulations have been shown to often agree well with experiments[42–46]; moreover, cavity photons are harmonic oscillators, whose dynamics can usually be captured well by classical dynamics[45]. By combining the classical motion of cavity photons and molecules, CavMD is expected to capture the essential physics in many VSC experiments. Second, CavMD has already correctly captured some key features of both equilibrium and nonequilibrium VSC experiments[23,38,47,48], including (i) the asymmetry of a Rabi splitting[28,47], (ii) polaritonic relaxation to vibrational dark modes on a timescale of ps or sub-ps[24,38,48], (iii) polariton-enhanced molecular nonlinear absorption[24]—a phenomenon whose signature is a delayed population gain in the first excited state of vibrational dark modes after strongly exciting the LP[7,23], and (iv) dark-mode relaxation dynamics for systems with large cavity volumes[34,38]. Altogether, classical CavMD appears to be a suitable choice between accuracy and affordability for simulating nonequilibrium vibrational dynamics under VSC.

The fundamental equations underlying CavMD and all simulation details for exciting a polariton in a liquid carbon dioxide system (including the force field) can be found in refs. [24,28]; see also Supplementary Methods for a comprehensive explanation. In short, as shown in Fig. 1, we imagine $N_{sub}$ molecules in a periodic simulation cell coupled to a single cavity mode (with two polarization directions). The effective coupling strength between each molecule and the cavity mode is denoted as $\widetilde{\varepsilon}$, which is defined as

$$\widetilde{\varepsilon} \equiv \sqrt{N_{cell}} \varepsilon. \tag{1}$$

Here, $\varepsilon = \sqrt{m_c \omega_c^2 / \Omega \epsilon_0}$ denotes the true light–matter coupling strength between each molecule and the cavity mode, where $\omega_c$, $\Omega$ and $\epsilon_0$ denote the cavity mode frequency, the effective cavity volume, and the vacuum permittivity, respectively, and $m_c = 1$ a.u. denotes the auxiliary mass for the cavity photons (see Supplementary Methods for details). $N_{cell}$ denotes the number of periodic simulation cells, the value of which can be determined by fitting the experimentally observed Rabi splitting. As discussed in Supplementary Methods, when CavMD is used to simulate VSC in Fabry–Pérot microcavities, we must always check that our results on a small subsystem are not spurious. We check for this possibility by running a protocol that enlarges the molecular system size (while fixing the Rabi splitting and molecular density)—the results in the asymptotic limit of a very large $N_{sub}$ correspond to the true experimental result. See Fig. 4 for such a system size enlarging process.

For the simulations reported here, by using an explicit force field defined in Supplementary Methods, we include vibrational relaxation and dephasing for the molecular subsystem. We have assumed that the molecular force field remains the same inside versus outside the cavity, i.e., both intramolecular and intermolecular interactions are unchanged under VSC. For nonpolar molecules, although a recent first-principle calculation shows that intermolecular interactions can be modified by several cm$^{-1}$ when a few molecules are confined in a cavity[49], we anticipate that, compared with thermal fluctuations at 300 K, such a small modification can be neglected when the molecular number is large. We disregard cavity loss in the manuscript, which should usually be reasonable for Fabry–Pérot microcavities. Note that the main finding of this manuscript—selective polaritonic energy transfer to solute molecules—occurs mostly within 1 ps after the laser pumping, while a typical cavity mode lifetime for a Fabry–Pérot microcavity is ~5 ps[23]. In Supplementary Note 2 we have verified that a cavity lifetime larger than 1 ps would not meaningfully alter the results corresponding to Fabry–Pérot microcavities (i.e., when $N_{sub}$ is large). The code for reproducing this work, which is implemented based on the I–PI interference[50] and LAMMPS[51], is open-source and available at Github[52].

## Data availability

All the data shown here, including the script files to generate the raw molecular dynamics trajectories and the data for plotting, are available at https://github.com/TaoELi/cavity-md-ipi[52].

## Code availability

The code for reproducing this paper, including a tutorial for installation and simulations, is available at https://github.com/TaoELi/cavity-md-ipi[52].

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

## Acknowledgements

This material is based upon work supported by the U.S. National Science Foundation under Grant No. CHE1953701 (A.N.); and US Department of Energy, Office of Science, Basic Energy Sciences, Chemical Sciences, Geosciences, and Biosciences Division under Award No. DE-SC0019397 (J.E.S.).

## Author contributions

T.E.L. conceived the project, performed simulations and plotted the figures, T.E.L., A.N., and J.E.S. discussed the results and wrote the paper.

## Competing interests

The authors declare no competing interests.
