## [Peer Review File · Nature Communications]

REVIEWER COMMENTS

Reviewer #4 (Remarks to the Author):

The paper by Li et al. is an excellent piece of work and is highly relevant for the current discussions in polaritonic chemistry. The possibility to excite solute molecules using solvent polaritonic states is really exciting and is not limited only to molecular vibrations.

I have carefully read the paper together with the author's replies to the reviewer's comments. I find they have addressed all the issues raised by the reviewers in a satisfactory way. There seems to be some reservation among the reviewers as this is a theoretical paper making predictions about phenomena that is still to be confirmed experimentally. This seems unjustified as theoretical predictions are essential for advancing the experimental practice of polaritonic chemistry.

I, therefore, recommend the paper be published in Nature Communications after the authors have addressed the following minor points.

1. I find the text is in some places too wordy and I personally would appreciate that authors getting to the point faster.
2. In the caption to fig. 1 it is stated "with two polariton directions" – I have no idea what this is, should this be the two polarizations? – also if two polarizations are considered then these are two-photon modes as the photons are different.
3. The employed model is rather crude, but I think it is sufficiently accurate to make the predictions being proposed. However, it has recently been shown by Haugland et al. (JCP 154, 094113, 2021)

that cavities can alter van der Waals interactions among non-polar molecules in cavities. I find it appropriate that the authors mention this work and discuss the implications for the force field used in the reported study.

4. I find the graphics in the paper rather poor and I would have expected the authors to have put more energy into preparing the figures, especially when aiming for high-impact journals such as Nature Chemistry and Nature Communications. I am not really sure that the cartoons in fig 1 illustrate the message of the paper that well – seems “right” should be left? In fig 2 I would have preferred to reverse that color scale making it much easier to read – also white text on a black background is not easy to read. I would probably find it useful to have the “quantum picture” figure in the main text and not in the SI.

5. It would be very useful if the authors could include the Fourier transform of the external pulse to see the frequency broadening and shape.

The comments of the authors are reported in blue.

The authors propose a pumping scheme to selectively populate highly-excited vibrations ($v=2$) in liquid phase, where a small number ($N=10$) of solute molecules ($^{13}\text{CO}_2$) are surrounded by a way larger number ($N=206$) of solvent molecules ($^{12}\text{CO}_2$). The pumping scheme makes use of a cavity — resonant with the solvent vibrations — and an intense infrared laser. Via a highly non-linear excitation, the authors claim that the solvent molecules transfer energy directly to the solute aided by the collective strong coupling. As a result, the pumping scheme defies the normal entropic effect, which would result in the energy being stored mostly in the solvent with a subsequent heating of the system.

The authors investigate the system via classical simulations of cavity molecular dynamics (CavMD), including explicit pumping with a strong laser and investigating the effects of different lossy channels, accounted both at the individual and at the global level. The paper is very well-written and it presents a very powerful scheme, with a large potential to open up a pathway to IR-photochemistry in liquid phase, which nowadays is extremely difficult to control due to the heating effect mentioned above.

Judging from the remarkable results and the potential applications point of view, the paper is extremely interesting and I believe it would constitute a very relevant piece of literature to stimulate discussions and experiments. However, the underlying mechanism leading to the excitation of highly-excited states of the solute has been the object of several controversies in the history of the manuscript, which shall be taken into account before I can support the publication in *Nature Communications*.

Going into the detail: two main points have been raised and argued, the second of which I personally consider more puzzling.

- The first one is the feasibility of the method to describe such a complex pumping scheme without experimental validation, arguing that the model lacks many effects such as local field on the solvent.

While I see the point of the argument that local field effects on solvents in cavity might be relevant, it is also true that an extensive exploration of this topic should deserve a standalone work. Also, as frequently happens even with the most refined models, a qualitative agreement in such disordered and complex systems is the best that can potentially be achieved. The point of publishing a theoretical work such as the one presented here is to provide a hint for experiments to explore this effect in similar setups (opportunistically modified/updated). Within this paradigm, I believe the validation of the cavMD method on similar systems presented in previous works is enough to prove the point of a qualitative proposal. To summarise, I believe it would be interesting to see (as a theoretical curiosity) how local field effects might impact the proposed mechanism, yet not really life-changing for the qualitative message of the present work.

- The second one is about how trivial the underlying pumping mechanism is. In particular, it has been argued that the lower polaritonic mode formed between cavity and solvent acts as a slightly modified effective cavity mode. The final effect of this effective cavity mode is that it enhances nonlinear absorption of a transition that is resonant with the solute. Hence, the main point of the criticism is that the mechanism leading to highly-excited vibration of solute molecules is NOT an energy transfer process (from solvent to solute) induced by vibrational strong coupling. Instead, it is more like a multiphoton nonlinear pumping of the solute via an effective cavity mode, which is a well-known effect.

The authors have shown that indeed, if the cavity was filled *only* with solute molecules, a non-linear pumping via a slightly detuned cavity mode would achieve the same effect in weak coupling, observing practically the same energy storage in the solute molecules. While this is an undeniable fact, the two compared systems are not exactly comparable as the authors state:

“As we replied in the second response letter (response_round2.pdf), although Fig. R2d does show the strong excitation of the $^{13}\text{CO}_2$ molecules, this strong excitation simply comes from the fact that there are no $^{12}\text{CO}_2$ solvent molecules, so there are no other molecules (apart from a few $^{13}\text{CO}_2$ molecules) to absorb energy from the excited cavity mode. That being said, and this where we strongly disagreed with the reviewer, our goal in this research manuscript is to argue that, if one lines up energy levels correctly, one can use a solvent polariton to excite an impurity even in the presence of solvent. In other words, while one approach might work in vacuum, one must recognize that very different approaches are necessary in the condensed phase where different physics manifests itself.”

That being said, I do not believe that the evidence presented so far by the authors is the best to prove unambiguously that the mechanism is due to energy transfer processes, in particular I am referring to Figure S7. I have been trying to think about additional evidence that the authors may produce to dissipate any doubt about the mechanism, however I agree with the authors that *“it may be difficult to assess exactly how much of the interaction between polariton and impurity is dictated by the photon-impurity vs solvent-impurity interactions (because disentangling the photon from the solvent components of a polariton may not be well-defined)”*:

1) Associated to the energy plots they present, I would suggest that the authors show an explicit dynamics where they track the time evolution of the different intermolecular interaction channels (solute-solvent, solvent-cavity, cavity-solute).

2) I suggest they present also individual simulations where they selectively "turn off" selected forces, with the final goal of singling out the effect of each interaction channel in the dynamics (a useful examples would be the cavity-free intermolecular forces and the cavity-free solute-solvent forces). If the energy storage is indeed an effect of the dephasing as the authors state: *“what is clear is that localization of energy on a few molecules does require dephasing, a many-body effect that arises only from molecule-molecule interactions.”* then the pumping to the solvent would be dominant and the transfer process should not occur. However, if the driving is a simple effect of multiphoton pumping depending on the energetics, there should be relatively few differences in the energy stored by the solute.

3) Track the dynamics of the energy stored in the cavity (photons), to verify whether the energy stored in the solvent is directly injected in the solute or if the cavity mediates the process.

4) Reducing the cavity lifetime to something extremely short after the excitation pulse, proving how relevant the residual energy stored in the cavity is to keep driving the solute modes.

Response to Reviewers

We thank the three reviewers for their helpful comments and suggestions. We have responded to all of their comments below and have revised the manuscript accordingly. The original comments of the referees are in black, our responses are in blue, and our modifications of the manuscript are in red as quoted below.

Reviewer #4 (Comments for the Author):

The paper by Li et al. is an excellent piece of work and is highly relevant for the current discussions in polaritonic chemistry. The possibility to excite solute molecules using solvent polaritonic states is really exciting and is not limited only to molecular vibrations.

I have carefully read the paper together with the author's replies to the reviewer's comments. I find they have addressed all the issues raised by the reviewers in a satisfactory way. There seems to be some reservation among the reviewers as this is a theoretical paper making predictions about phenomena that is still to be confirmed experimentally. This seems unjustified as theoretical predictions are essential for advancing the experimental practice of polaritonic chemistry.

I, therefore, recommend the paper be published in Nature Communications after the authors have addressed the following minor points.

We thank the reviewer for the positive assessment and the helpful comments. We address each comment below.

1. I find the text is in some places too wordy and I personally would appreciate that authors getting to the point faster.

We have deleted some sentences in the introduction and have also moved the introduction of CavMD to Sec. Methods to avoid being very wordy.

2. In the caption to fig. 1 it is stated "with two polariton directions" – I have no idea what this is, should this be the two polarizations? – also if two polarizations are considered then these are two-photon modes as the photons are different.

Sorry for this typo. In Fig. 1, we have changed "with two polariton directions" to "with two polarization directions".

For the reviewer's comment that "if two polarizations are considered then these are two-photon modes as the photons are different", we apologize if our nomenclature was unclear. As stated in the manuscript, we refer to our cavity system as "a single cavity mode" because the photonic energies for the two polarization directions are the same, and only a single photon peak can appear in the IR spectroscopy, so there is only a single mode from a spectroscopic point of view. Of course, we could also refer to our cavity system as "two energy-degenerated cavity photons with different polarization directions", but our feeling is that this approach would confuse many

readers because in many theory papers of cavity polaritons, only a single cavity mode with a single polarization direction is used.

We have added some discussion in SI page 9 to make this point clear:

“As stated in the manuscript and above, we refer to our cavity system as ‘a single cavity mode’ because the photonic energies for the two polarization directions are the same, and only a single photon peak appears in the IR spectroscopy -- so there is only a single mode *from a spectroscopic point of view*. Of course, we could also refer to our cavity system as a system with ‘two energy-degenerated cavity photons with different polarization directions’, but this description seems a bit lengthy.”

3. The employed model is rather crude, but I think it is sufficiently accurate to make the predictions being proposed. However, it has recently been shown by Haugland et al. (JCP 154, 094113, 2021) that cavities can alter van der Waals interactions among non-polar molecules in cavities. I find it appropriate that the authors mention this work and discuss the implications for the force field used in the reported study.

We thank the reviewer a lot for pointing out this useful reference! In Methods of the manuscript (page 8), we have mentioned this work and its relationship with the force field:

“We have assumed that the molecular force field remains the same inside versus outside the cavity, i.e., both intramolecular and intermolecular interactions are unchanged under VSC. For nonpolar molecules, although a recent first-principle calculation shows that intermolecular interactions can be modified by several cm^{-1} when a few molecules are confined in a cavity \cite{Haugland2021}, we anticipate that, compared with thermal fluctuations at 300 K, such a small modification can be neglected when the molecular number is large.”

4. I find the graphics in the paper rather poor and I would have expected the authors to have put more energy into preparing the figures, especially when aiming for high-impact journals such as Nature Chemistry and Nature Communications. I am not really sure that the cartoons in fig 1 illustrate the message of the paper that well – seems “right” should be left? In fig 2 I would have preferred to reverse that color scale making it much easier to read – also white text on a black background is not easy to read. I would probably find it useful to have the “quantum picture” figure in the main text and not in the SI.

We have modified the cartoon in Fig. 1 and added the “quantum picture” in Fig. 1 for a better illustration.

For Fig. 2, we have taken the reviewer’s advice and reversed the color.

5. It would be very useful if the authors could include the Fourier transform of the external pulse to see the frequency broadening and shape.

The Fourier transform of the external pulse is now shown in Fig. 2.

Reviewer #5 (Comments for the Author):

The authors propose a pumping scheme to selectively populate highly-excited vibrations ($v=2$) in liquid phase, where a small number ($N=10$) of solute molecules (13CO_2) are surrounded by a way larger number ($N=206$) of solvent molecules (12CO_2). The pumping scheme makes use of a cavity — resonant with the solvent vibrations — and an intense infrared laser. Via a highly non-linear excitation, the authors claim that the solvent molecules transfer energy directly to the solute aided by the collective strong coupling. As a result, the pumping scheme defies the normal entropic effect, which would result in the energy being stored mostly in the solvent with a subsequent heating of the system.

The authors investigate the system via classical simulations of cavity molecular dynamics (CavMD), including explicit pumping with a strong laser and investigating the effects of different lossy channels, accounted both at the individual and at the global level. The paper is very well-written and it presents a very powerful scheme, with a large potential to open up a pathway to IR-photochemistry in liquid phase, which nowadays is extremely difficult to control due to the heating effect mentioned above.

Judging from the remarkable results and the potential applications point of view, the paper is extremely interesting and I believe it would constitute a very relevant piece of literature to stimulate discussions and experiments. However, the underlying mechanism leading to the excitation of highly-excited states of the solute has been the object of several controversies in the history of the manuscript, which shall be taken into account before I can support the publication in Nature Communications.

We thank the reviewer for the positive assessment and the helpful comments. We address each comment below.

Going into the detail: two main points have been raised and argued, the second of which I personally consider more puzzling.

- The first one is the feasibility of the method to describe such a complex pumping scheme without experimental validation, arguing that the model lacks many effects such as local field on the solvent.

While I see the point of the argument that local field effects on solvents in cavity might be relevant, it is also true that an extensive exploration of this topic should deserve a standalone work. Also, as frequently happens even with the most refined models, a qualitative agreement in such disordered and complex systems is the best that can potentially be achieved. The point of publishing a theoretical work such as the one presented here is to provide a hint for experiments to explore this effect in similar setups (opportunistically modified/updated). Within this paradigm, I believe the validation of the CavMD method on similar systems presented in previous works is enough to prove the point of a qualitative proposal. To summarise, I believe it would be interesting to see (as a theoretical curiosity) how local field effects might impact the proposed mechanism, yet not really life-changing for the qualitative message of the present work.

We agree with the reviewer's comments that the current force-field treatment should be OK to "prove the point of a qualitative proposal". The authors will report more accurate CavMD simulations of collective vibrational strong coupling beyond classical force fields in a separated work.

- The second one is about how trivial the underlying the pumping mechanism is. In particular, it has been argued that the lower polaritonic mode formed between cavity and solvent acts as a slightly modified effective cavity mode. The final effect of this effective cavity mode is that it enhances nonlinear absorption of a transition that is resonant with the solute. Hence, the main point of the criticism is that the mechanism leading to highly-excited vibration of solute molecules is NOT an energy transfer process (from solvent to solute) induced by vibrational strong coupling. Instead, it is more like a multiphoton nonlinear pumping of the solute via an effective cavity mode, which is a well-known effect.

The authors have shown that indeed, if the cavity was filled only with solute molecules, a non-linear pumping via a slightly detuned cavity mode would achieve the same effect in weak coupling, observing practically the same energy storage in the solute molecules. While this is an undeniable fact, the two compared systems are not exactly comparable as the authors state:

"As we replied in the second response letter (response_round2.pdf), although Fig. R2d does show the strong excitation of the $^{13}\text{CO}_2$ molecules, this strong excitation simply comes from the fact that there are no $^{12}\text{CO}_2$ solvent molecules, so there are no other molecules (apart from a few $^{13}\text{CO}_2$ molecules) to absorb energy from the excited cavity mode. That being said, and this where we strongly disagreed with the reviewer, our goal in this research manuscript is to argue that, if one lines up energy levels correctly, one can use a solvent polariton to excite an impurity even in the presence of solvent. In other words, while one approach might work in vacuum, one must recognize that very different approaches are necessary in the condensed phase where different physics manifests itself."

That being said, I do not believe that the evidence presented so far by the authors is the best to prove unambiguously that the mechanism is due to energy transfer processes, in particular I am referring to Figure S7. I have been trying to think about additional evidence that the authors may produce to dissipate any doubt about the mechanism, however I agree with the authors that "it may be difficult to assess exactly how much of the interaction between polariton and impurity is dictated by the photon-impurity vs solvent-impurity interactions (because disentangling the photon from the solvent components of a polariton may not be well-defined)".

1) Associated to the energy plots they present, I would suggest that the authors show an explicit dynamics where they track the time evolution of the different intermolecular interaction channels (solute-solvent, solvent-cavity, cavity-solute).

We sincerely thank the reviewer for the helpful suggestions. Detailed simulation data are presented below.

2) I suggest they present also individual simulations where they selectively "turn off" selected forces, with the final goal of singling out the effect of each interaction channel in the dynamics (a useful examples would be the cavity-free intermolecular forces and the cavity-free solute-solvent forces). If the energy storage is indeed an effect of the dephasing as the authors state: "what is clear is that localization of energy on a few molecules does require dephasing, a many-body effect that arises only from molecule-molecule interactions." then the pumping to the solvent would be dominant and the transfer process should not occur. However, if the driving is a simple effect of multiphoton pumping depending on the energetics, there should be relatively few differences in the energy stored by the solute.

Figure S3. (a) Similar plot as Fig. S2(b) except that the pulse fluence is reduced from 632 mJ/cm^2 to 158 mJ/cm^2 . Under a smaller pulse fluence, after the pulse excitation (the yellow window), the energy transfer from the $^{12}\text{CO}_2$ solvent to the $^{13}\text{CO}_2$ solute species can be identified more easily. (b) Replot of Fig. (a) except that the intermolecular interactions involving $^{13}\text{CO}_2$ molecules are turned off. In the absence of the $^{13}\text{CO}_2$ intermolecular interactions, the $^{12}\text{CO}_2 \rightarrow ^{13}\text{CO}_2$ intermolecular vibrational energy transfer is greatly suppressed.

As suggested by the reviewer, in Fig. S3, we have performed two additional simulations to prove our point that “the mechanism leading to highly-excited vibration of solute molecules” is “an energy transfer process (from solvent to solute) induced by vibrational strong coupling” and is not a simple “multiphoton nonlinear pumping of the solute via an effective cavity mode.”

In Fig. S3, we plot the total potential energy *per species*: $^{12}\text{CO}_2$ solvent (gray line), $^{13}\text{CO}_2$ (red line), and the cavity mode (cyan line) under two different conditions.

For the first simulation (to be compared against the simulation condition in Fig. 3b of the manuscript (or Fig. S2b)), in Fig. S3a we have reduced the pulse fluence from $F = 632 \text{ mJ/cm}^2$ to $F = 158 \text{ mJ/cm}^2$. Such a reduction of the pulse fluence can increase the polariton lifetime and give better time-resolved energy transfer dynamics than that in Fig. S2b. In Fig. S3a, after the pulse excitation (yellow window), we find that both the solvent and photonic energies are transferred to the solute molecules, showing that forming solvent polaritons can indeed facilitate intermolecular vibrational energy transfer. Although the $^{13}\text{CO}_2$ solute concentration is only 4.63%, most the input energy is accumulated in the solute molecules at $t = 3 \text{ ps}$.

For the second simulation, we further remove the intermolecular interactions involving the $^{13}\text{CO}_2$ solute species. In other words, Fig. S3b is an *unphysical* simulation where the intermolecular

interactions between each $^{13}\text{CO}_2$ and other molecules ($^{12}\text{CO}_2$ or $^{13}\text{CO}_2$) are artificially turned off. Although Fig. S3b is unphysical, this plot shows unambiguously that, as compared with Fig. S3a, the energy transfer from the $^{12}\text{CO}_2$ species to the $^{13}\text{CO}_2$ species is greatly suppressed, demonstrating the importance of intermolecular interactions in facilitating energy transfer. Overall, Fig. S3b shows that our mechanism of energy accumulation in the solute species after the solvent polariton pumping is indeed a many-body mechanism and intermolecular interactions play an important role.

The above two paragraphs have been added to the SI around Fig. S3. We thank the reviewer for pushing us on this point and offering a suggestive criticism; the paper is undoubtedly more convincing now.

3) Track the dynamics of the energy stored in the cavity (photons), to verify whether the energy stored in the solvent is directly injected in the solute or if the cavity mediates the process.

In SI Figs. S2 and S3, we have plotted the photonic energy dynamics as the cyan lines. The discussion of Fig. S3 is given above.

In fairness, it is impossible to “*verify whether the energy stored in the solvent is directly injected in the solute or if the cavity mediates the process*” with absolutely certainty. The difficulty arises because the solvent molecules are strongly coupled to the cavity mode, and the coherent energy transfer timescale between the solvent bright mode and the cavity mode is faster than the timescale of the energy transfer to the solute molecules. Due to such a fast coherent energy transfer, we cannot simply separate the effect of the solvent or the cavity modes vis a vis the polariton decay. That being said, the data above show unambiguously that the intermolecular interactions between solvent and solute are an essential component of this energy transfer--the energy transfer is NOT facilitated exclusively through the photon--which does refute the earlier referee's criticism.

4) Reducing the cavity lifetime to something extremely short after the excitation pulse, proving how relevant the residual energy stored in the cavity is to keep driving the solute modes.

In SI Fig. S4 (Fig. S3 in the last version), we have shown the effect of cavity loss. Given the same overall Rabi splitting, when the molecular number is large enough, cavity loss has nearly no effect of solvent-polariton energy transfer to the solute molecules --- even when the cavity loss is very large, e.g., a 0.3 ps cavity lifetime.

Reviewer #6 (Comments for the Author):

The phenomenon described in the manuscript, discovered by computational experiments, has in my opinion the qualities that recommend publication in Nature Communications. It is unexpected, relates to a rapidly developing field such as (collective) molecule-light strong coupling, its essence is involving basic concepts accessible to a broad readership. What I find unsatisfactory is the rationalization of the effect. In fact, even if the underlying computational model is based on classical atoms, the explanation is provided making reference to a quantum mechanical picture (the resonance between twice the polaritonic energy and the 0-2 transitions of

the solute molecule, with the polariton acting as a virtual state for the excitation). Although reading ref. 22 I understand the relation drawn between vibrational frequencies of anharmonic oscillators and vibrational quanta occupation, invoking virtual states in a classical picture seems to imply the model contains more than it really does. My (fully classical) understanding of what happens is as follows: once the solvent LP is excited, it starts to drive the oscillation of the solute molecules -despite the detuning of the linear absorption frequencies- via the local oscillating electric field associated to the excited (classical) LP. The more the solute molecules absorb energy by this non-resonant but strong driving field, the more their vibrational frequency downshifts, tuning with the LP one, "self-catalizing" further absorption till complete energy transfer.

In short, I am certainly in favor of publication of this work, but I think the explanation of the effect should be clearly given in classical terms (perhaps not what I sketched here above, the authors may well have alternative/more detailed explanations) to be convincing.

We thank the reviewer for the positive assessment and the helpful comments. We have modified the manuscript to add the classical interpretation of the polariton-enhanced molecular nonlinear absorption mechanism (see page 3 left column of the manuscript).

“This quantum-mechanical interpretation of polariton-enhanced molecular nonlinear absorption is easy to understand. A classical analog can be pictured as well. Once the LP is excited and starts to dephase, due to the inhomogeneous local molecular environment, some molecules receive more polaritonic energy than the others. Due to molecular anharmonicity, these high-energy molecules oscillate with red-shifted, closer-to-LP frequencies, leading to an even stronger interaction between these high-energy molecules and the LP. Such "self-catalyzing" can, under strong LP excitation, eventually lead to some molecules being strongly and nonlinearly excited. While keeping in mind that our calculations are in fact classical, below we will use the quantum description because we feel the latter is more standard in the literature.”

We thank all the three reviewers for their helpful suggestions and hope that this revised version is now suitable for publication in Nature Communications.

REVIEWERS' COMMENTS

Reviewer #5 (Remarks to the Author):

In the revised version of the manuscript, the authors present convincing evidence that the mechanism they propose is indeed a cavity-assisted vibrational energy transfer from the solvent to the solute, and not a conventional multi-photon nonlinear absorption. To prove their point, the authors artificially removed the intermolecular interactions, showing a large quenching in the energy accumulated in the solute. Although a system where intermolecular interactions are turned off is unphysical, this simulation proves the key role of intermolecular interactions in accumulating energy in the solute's vibrational modes. The nature of this mechanism has been probably the most controversial point in the history of the manuscript. I believe that the revised version of the manuscript shows unambiguously how the strong coupling regime and the intermolecular interactions are both necessary to achieve the selective excitation of solute molecules embedded in a solvent, hence solving the controversy on the mechanism. Therefore, I strongly support the publication of the current version of the manuscript in Nature Communications without further revisions.

Response to Reviewers

We thank reviewer #5 for their helpful comments and suggestions. We have responded to all of their comments below and have revised the manuscript accordingly. The original comments of the referees are in black, our responses are in blue, and our modifications of the manuscript are in red as quoted below.

Reviewer #5 (Comments for the Author):

In the revised version of the manuscript, the authors present convincing evidence that the mechanism they propose is indeed a cavity-assisted vibrational energy transfer from the solvent to the solute, and not a conventional multi-photon nonlinear absorption. To prove their point, the authors artificially removed the intermolecular interactions, showing a large quenching in the energy accumulated in the solute. Although a system where intermolecular interactions are turned off is unphysical, this simulation proves the key role of intermolecular interactions in accumulating energy in the solute's vibrational modes. The nature of this mechanism has been probably the most controversial point in the history of the manuscript. I believe that the revised version of the manuscript shows unambiguously how the strong coupling regime and the intermolecular interactions are both necessary to achieve the selective excitation of solute molecules embedded in a solvent, hence solving the controversy on the mechanism. Therefore, I strongly support the publication of the current version of the manuscript in Nature Communications without further revisions.

We thank the reviewer for the positive assessment and the supporting the publication without further revisions.